# SLR: Learning Quadruped Locomotion without Privileged Information

Shiyi Chen[1], Zeyu Wan[1], Shiyang Yan[1], Chun Zhang[*,1], Weiyi Zhang[1], Qiang Li[*,2], Debing Zhang[1], Fasih Ud Din Farrukh[1]

[1]Tsinghua University, [2]Shenzhen Technology University

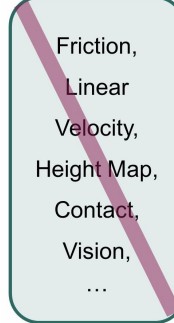 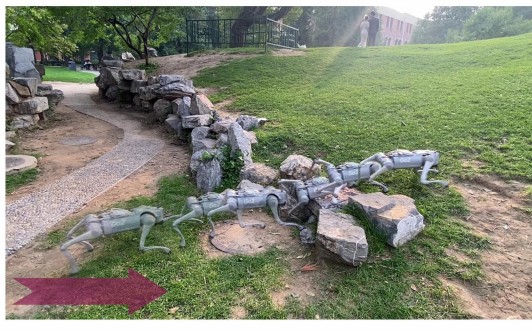

Figure 1: We propose a framework for training a robust quadruped locomotion policy without relying on privileged information. The robot effectively navigates challenging terrains, showcasing adaptive locomotion skills acquired through self-learning.

**Abstract:** The recent mainstream reinforcement learning control for quadruped robots often relies on privileged information, demanding meticulous selection and precise estimation, thereby imposing constraints on the development process. This work proposes a Self-learning Latent Representation (SLR) method, which achieves high-performance control policy learning without the need for privileged information. To enhance the credibility of the proposed method's evaluation, SLR was directly compared with state-of-the-art algorithms using their open-source code repositories and original configuration parameters. Remarkably, SLR surpasses the performance of previous methods using only limited proprioceptive data, demonstrating significant potential for future applications. Ultimately, the trained policy and encoder empower the quadruped robot to traverse various challenging terrains. Videos of our results can be found on our website: https://11chens.github.io/SLR/

**Keywords:** Locomotion, Reinforcement Learning, Privileged Learning

## 1 Introduction

Humans and animals inherently possess locomotion abilities, enabling them to traverse various complex terrains. In contrast, gait control for robots is highly challenging. Model-based methods

---

[*]Corresponding Author

8th Conference on Robot Learning (CoRL 2024), Munich, Germany.

have achieved some success by leveraging robots' mechanical structures and dynamic principles [1, 2, 3, 4, 5]. However, finding a balance between model accuracy and computational efficiency remains difficult, especially for real-time applications.

Additionally, designing these models requires a deep understanding of a robot dynamics, posing a significant challenge for researchers. As a result, Reinforcement Learning (RL) methods are becoming increasingly popular. By simulating real-world environments and training policies with customized reward functions, these methods enable robots to perform complex locomotion tasks in real-time [6, 7, 8, 9, 10, 11, 12, 13, 14, 15, 16, 17, 18].

The recent mainstream RL applications in quadruped robots rely on privileged learning methods [19]. In real-world scenarios, a robot's interaction with its environment is modeled as a Partially Observable Markov Decision Process (POMDP). Solely relying on proprioceptive sensor measurements, a robot cannot fully perceive external environmental information, limiting its decision-making capabilities. Consequently, many studies leverage the "observability" advantages of simulation platforms. During training, various physical parameters (such as friction coefficients [10, 20, 21], restitution coefficients [9, 21], and scan-dots of the terrain [14, 22]) are artificially added as privileged information to help the robot understand both itself and the external environment.

Unlike human cognition, which navigates terrains without explicit knowledge of physical parameters, neural network-based robots may not benefit from adding such parameters. This is because the parameters are not used to solve dynamic equations but are simply input into the neural network. As a result, these privileged information pieces may not enhance the interpretability or effectiveness of neural network-based agents.

Therefore, instead of relying on manually chosen physical parameters to construct privileged information, this work explores whether it is possible for robots to learn a latent representation of environmental states by themselves? It is proposed that intelligent robots should be capable of self-learning latent representations, which are inherently more suitable than those predefined by humans.

To achieve this, we propose the Self-learning Latent Representation (SLR) algorithm, which generates latent representations guided by a RL Markov process without privileged information. This self-learning approach enables the robot to grasp latent environmental features and exhibit generalized locomotion capabilities across challenging terrain, as illustrated in Figure 1.

Our results show that the SLR algorithm, which operates without privileged information, outperforms traditional privileged learning methods. It consistently aligns with actual terrain conditions across various terrains. Implemented in existing open-source repositories and evaluated in the same environments as previous studies, the SLR algorithm demonstrates state-of-the-art (SOTA) performance in both simulations and real-world applications. This suggests that the self-learning approach has considerable potential for future expansion.

## 2 Related Work

Privileged learning in RL-based methods can be divided into explicit estimation and implicit estimation methods based on the target of supervised learning. In this paper, directly fitting specific physical parameters from privileged information is referred to as explicit estimation, while fitting the latent representation of privileged information is classified as implicit estimation.

**Explicit Estimation.** [6] concurrently trains a policy network and a state estimator, which include real-world parameters that are difficult to obtain accurately, such as linear velocity, foot height, and contact probability. [13] sets friction coefficients and stiffness coefficients as privileged information, inferring these from observation history to assist the robot in domain randomizations. But in the real world, the quadruped robot's foot contact time is very short during fast running, making it difficult to fully perceive ground friction. Therefore, [12] proposes learning information-gathering behaviors by adding an active estimation reward, which increases the accuracy of estimates for privileged information disparities.

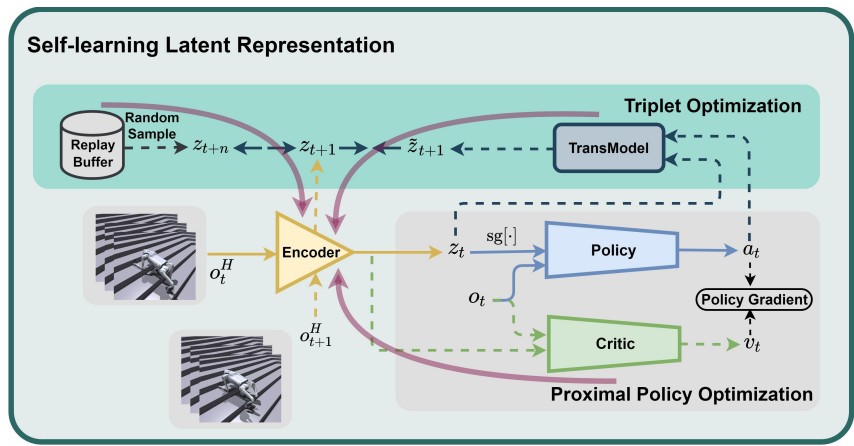

Figure 2: Illustration of SLR training framework. All dashed lines represent the network updating process. The translucent fuchsia lines indicate the encoder updates through backpropagation from the Critic network, the Transition model, and random sampling. The remaining solid lines represent the network's forward inference process.

**Implicit Estimation.** To train quadruped robots on complex terrain, estimating a large amount of privileged information explicitly is challenging. A common approach is to encode this high-dimensional information into a latent representation. [10] utilizes a teacher encoder to compress data such as friction and terrain height into a latent representation, and then a student adaptation module learn to infer this latent representation from observation history. [11] optimized the teacher-student training process from [10] in one stage by regularizing the teacher's actual privileged latent and supervising the student's estimated privileged latent. The algorithm of [9] also includes a teacher-student policy. During training, the teacher's encoder is used, and the student's adaptation module output aligns with the teacher's encoder. For deployment, the student's adaptation module and the teacher's trained policy are utilized together. [8] employs the Asymmetric Actor-Critic (AAC) [7] method, feeding privileged information to the critic network while using encoder-decoder estimators to assist the actor in imagining this privileged information.

## 3 Method

### 3.1 Problem Formulation

Our goal is to construct a one-stage end-to-end system based on RL, using proprioceptive sensor data as input for measuring and controlling joint movements.

**Observation Space.** The observations $o_t$ consist of 45 dimensions, including the robot's base angular velocity $\omega$, commands from the joystick $c_t=[v_x^{\mathrm{cmd}}, v_x^{\mathrm{cmd}}, \omega_{\mathrm{yaw}}^{\mathrm{cmd}}]$, measurement of the gravity vector $g_t$, joint positions $\theta$ and velocities $\dot{\theta}$, and the actions $a_{t-1}$ taken by the robot at the previous time step.

**Action Space.** The action space is a 12-dimensional vector $a_t$ corresponding to the four legs of the quadruped robot, with each leg having three motor drive units. The neural network's output is converted into actual torque $\tau$ through a PD controller.

**Reward Functions.** The reward functions we used during the training are shown in Table A1, which come from [10, 16, 23].

**Domain Randomizations.** For the training of our method, domain randomizations are used. The details are shown in Table A2, which come from [8, 21].

## 3.2 Framework Overview

The proposed training framework utilizes the Markov Decision Process (MDP) to guide self-learning of the latent representation through state transitions, distinctions, and cumulative rewards, without relying on manually defined privileged information. All networks in this framework are multi-layer perceptrons (MLPs). Table A3 in the Appendix details the MLP structures, and Table A4 lists the training hyperparameters. The training framework is illustrated in Figure 2.

**Encoder:** In this architecture, the encoder's input consists of the observation history $o_t^H$, which is composed of proprioceptive information $o_t$ from the previous 10 time steps. The encoder outputs a 20-dimensional latent representation $z_t$ of the observation history:

$$z_t = \phi(o_t^H) \tag{1}$$

**Actor-Critic:** The policy (Actor) and Critic network are trained jointly via Proximal Policy Optimization (PPO) algorithm [24]. The policy takes as input the current proprioceptive observation $o_t$ and the latent representation $z_t$, and it outputs the joint position $a_t$. The Critic network, which shares the same input as the policy, outputs the state value $v_t$. It is worth noting that, we turn off the gradient of $z_t$ in the policy and turn it on in the Critic network, using the backpropagation of the Critic network to update the encoder in the direction of the maximum cumulative reward:

$$a_t = \pi\left(o_t, \text{sg}[z_t]\right) \tag{2}$$

$$v_t = V\left(o_t, z_t\right) \tag{3}$$

where $\text{sg}[\cdot]$ is the stop gradient operator.

**Transition Model:** The Transition model simulates the real state transitions of the environment $p\left(s_{t+1} \mid s_t, a_t\right)$. Its input is the latent representation $z_t$ and the action $a_t$, and it outputs the next time step's latent estimation $\tilde{z}_{t+1}$:

$$\tilde{z}_{t+1} = \mu(z_t, a_t) \tag{4}$$

**Loss Function:** We align the estimated $\tilde{z}_{t+1}$ from the state transition model with the actual latent state $z_{t+1}$ at time $t+1$, while ensuring distinctiveness from other latent states $z_{t+n}$ at different times $t+n$. Drawing inspiration from [25, 26], we formulate a latent representation loss function utilizing a triplet loss, denoted as $\mathcal{L}_{\text{trip}}$.

$$\mathcal{L}_{\text{trip}}\left(z_{t+1}, \tilde{z}_{t+1}, z_{t+n}\right) = \max\left(\|z_{t+1} - \tilde{z}_{t+1}\|_2^2 - \|z_{t+1} - z_{t+n}\|_2^2 + m, 0\right), \quad \text{s.t.} \quad n \neq 1 \tag{5}$$

where $m$ is a margin that is enforced between $\tilde{z}_{t+1}$ and $z_{t+n}$ pairs, set to 1.0 in the experiment.

This updating strategy empowers the encoder to comprehend the dynamics of environmental state transitions and extract environmental attributes by assimilating state-action pairs derived from the MDP rollout. Finally, the triplet loss $\mathcal{L}_{\text{trip}}$ is multiplied by a triplet coefficient and added to the PPO loss $\mathcal{L}_{\text{ppo}}$ [24] for network updating. Details of simulation and training are in Appendix 1.

# 4 Experiments

## 4.1 Ablation Study for Latent Representations

To evaluate the proposed algorithm against traditional privileged learning algorithms, an ablation study was conducted. In this setup, commonly used privileged information $e_t \in \mathbb{R}^{10}$ was chosen, including parameters such as friction, restitution, foot height, and foot contact. The methods using different types of latent representations are as follows:

1) **Implicit:** The policy input includes the implicit privileged latent $l_t$, which is encoded from privileged information $e_t$ by adopting the teacher policy approach in [9, 10].

2) **Explicit:** The policy input includes the explicit privileged information $\tilde{e}_t = \phi(o_t^H)$.

3) **SLR w/ explicit:** The SLR policy input is the concatenation of the self-learning latent $z_t$ and explicit privileged latent $\tilde{e}_t$.

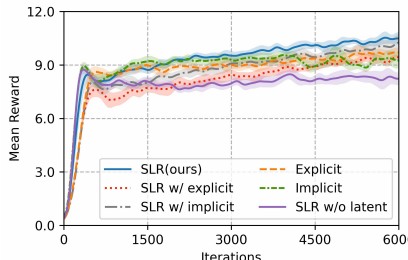
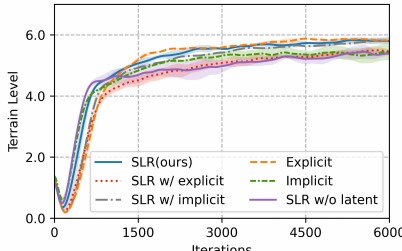

Figure 3: Ablation study training curves, curves are averaged over 3 seeds. The shaded area represents the standard deviation across seeds, and the curves are smoothed using Gaussian filtering.

4) **SLR w/ implicit:** The SLR policy input is the concatenation of the self-learning latent $z_t$ and implicit privileged latent $l_t$.

5) **SLR w/o latent:** The SLR policy input does not include any latent representations and does not utilize an encoder.

To ensure the fairness of quantitative comparisons, the configurations used in the experiments are all based on the default settings in the code repository [27]. Training was conducted for 6000 iterations under multiple terrains by default, and the results are presented in Figure 3. And we can draw the following conclusions:

1. The SLR algorithm outperforms traditional explicit and implicit privileged learning methods, suggesting that unified privileged information struggles to effectively represent diverse terrains.

2. Incorporating additional privileged latent information reduces the SLR algorithm's performance, possibly due to increased policy input disrupting the critic's guidance on updating the self-learning latent.

### 4.2 Latent Representation Analysis

Identifying and distinguishing terrain types is crucial for robotic systems [28]. To evaluate the self-learned latent representation, we conducted a simulation-based analysis. The test environment includes four terrains: an upward slope, descending stairs, flat ground, and ascending stairs. The robot navigates these terrains sequentially (Figure 4), and the latent representation output is recorded at each step. We then apply t-distributed stochastic neighbor embedding (t-SNE) to project these complex latent representations into a two-dimensional space for analysis. The latent representations from Implicit and SLR are examined separately, as shown in Figure 5.

**Different Terrain Representation.** As shown in Figure 5. The self-learned latent representations reveal distinct ring-shaped regions (labeled A, B, C, D) for each terrain. Notably, regions A and D exhibit similarity, suggesting the robot perceives up slope and ascending stairs as similar processes. In contrast, representations trained with the Implicit method present scattered points, indicative of overlapping privileged information across terrains.

**Terrain Transition Representation.** Notably, in the SLR latent representations, when the robot transitions from one terrain to another, all four ring-shaped latent representations show "tails". For instance, when transitioning from an up slope to descending stairs, the latent representation in region A has a tails extending to the right, with the tail's end having a color similar to the lower left corner of region B. This indicates that the robot is near the boundary between

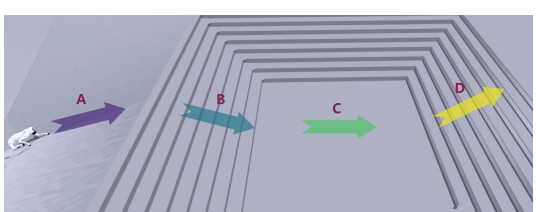

Figure 4: t-SNE test terrains.

terrains at that moment, and our latent representations are effectively indicating such terrain transitions.

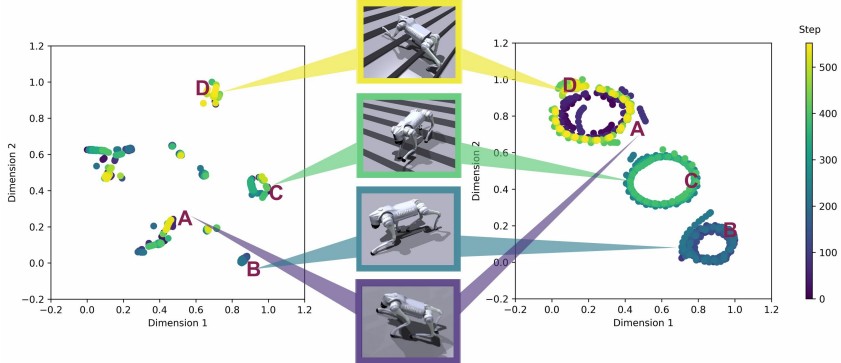

Figure 5: t-SNE visualization of Implicit (left) and SLR (right). Color intensity represents cumulative steps across four terrains. The privileged latent distribution is discrete and weakly correlated with terrain. In contrast, the SLR latent trajectories align precisely with the terrains traversed by the robot, with each ring-like representation accompanied by a "tail", indicating terrain transitions.

## 5 Results

### 5.1 Compared Methods

Previous comparative studies of robot algorithms often lack fairness due to the absence of a universal benchmark in robotics. Comparing algorithms within one's own environment is not objective, as reward functions and hyperparameters are typically optimized for the researcher's algorithm. Additionally, replicating previous algorithms in a new code repository can lead to incomplete reproductions, potentially affecting their performance.

In this study, these issues are addressed by directly implementing the SLR algorithm in the opensource code repositories of previous SOTA works [9, 11, 13, 27, 29]. Only the algorithm framework is modified while keeping other variables consistent with the original settings. This approach ensures a fair comparison between our method and the publicly available SOTA algorithms[1]. The algorithms which are evaluated as given below:

- **MoB**[13]: Explicit estimation, trained on flat ground with the Unitree Go1 robot.
- **RLvRL**[9]: Implicit estimation, trained on flat ground with the Cheetah robot.
- **ROA**[11]: Implicit estimation, trained on custom fractal noise environment using a Unitree Go1 robot equipped with a manipulator.
- **HIM**[29]: Explicit and implicit estimation, trained in multi-terrain environments with the Unitree Aliengo robot.
- **Baseline**[27]: No encoder, no privileged information, trained in multi-terrain environments with the Unitree A1 robot.

### 5.2 Simulation

All five code repositories implemented reinforcement learning based on the PPO algorithm, with training conducted on the IsaacGym platform [27, 30] using three different random seeds. The training curves are illustrated in Figure 6. Additionally, to evaluate the locomotion capabilities, experiments on velocity tracking performance were conducted on flat terrain. The evaluation metrics included Linear Velocity Tracking Error (LVTE) and Angular Velocity Tracking Error (AVTE), both measured using Mean Square Error (MSE). The evaluation results are summarized in Table 1.

---

[1]Previous evaluations in a single repository achieved noticeably better performance (see Section 4.1). To provide more conclusive evidence, we extended our experiments.

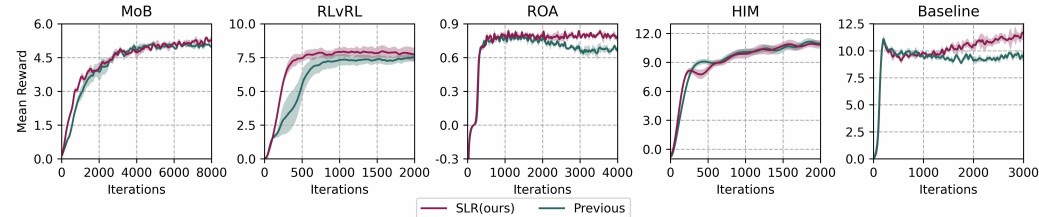

Figure 6: Training curves from various repositories show that the mean reward of SLR surpasses those of Baseline, RLvRL, and ROA, is slightly higher than MoB, and is comparable to HIM. The shaded area represents the standard deviation across seeds, and the curves are smoothed using Gaussian filtering. Note: SLR achieved a higher terrain level than HIM, indicating superior performance.

| Metrics | Ranges | Method | Code Repository | | | | |
|---------|--------|--------|-----------------|-----------------|-----------------|-----------------|-----------------|
| | | | MoB | RLvRL | ROA | HIM | Baseline |
| LVTE | [-1,1] m/s | Previous | $0.058^{\pm 0.011}$ | $0.032^{\pm 0.003}$ | $0.150^{\pm 0.001}$ | $0.014^{\pm 0.001}$ | $0.021^{\pm 0.005}$ |
| | | Ours | $\mathbf{0.029^{\pm 0.004}}$ | $\mathbf{0.024^{\pm 0.002}}$ | $\mathbf{0.136^{\pm 0.001}}$ | $\mathbf{0.013^{\pm 0.002}}$ | $\mathbf{0.011^{\pm 0.004}}$ |
| | [-2,2] m/s | Previous | $0.181^{\pm 0.014}$ | $0.108^{\pm 0.006}$ | N/A | $0.019^{\pm 0.009}$ | $0.074^{\pm 0.007}$ |
| | | Ours | $\mathbf{0.124^{\pm 0.023}}$ | $\mathbf{0.106^{\pm 0.002}}$ | N/A | $\mathbf{0.017^{\pm 0.002}}$ | $\mathbf{0.066^{\pm 0.004}}$ |
| AVTE | [-1,1] rad/s | Previous | $\mathbf{0.035^{\pm 0.002}}$ | $0.182^{\pm 0.007}$ | $1.100^{\pm 0.041}$ | $\mathbf{0.013^{\pm 0.009}}$ | $0.137^{\pm 0.013}$ |
| | | Ours | $0.049^{\pm 0.001}$ | $\mathbf{0.057^{\pm 0.007}}$ | $\mathbf{1.096^{\pm 0.025}}$ | $0.014^{\pm 0.011}$ | $\mathbf{0.077^{\pm 0.019}}$ |
| | [-2,2] rad/s | Previous | $0.131^{\pm 0.015}$ | $0.347^{\pm 0.002}$ | N/A | $0.038^{\pm 0.019}$ | $0.315^{\pm 0.021}$ |
| | | Ours | $\mathbf{0.086^{\pm 0.009}}$ | $\mathbf{0.142^{\pm 0.002}}$ | N/A | $\mathbf{0.038^{\pm 0.023}}$ | $\mathbf{0.223^{\pm 0.026}}$ |

Table 1: Velocity tracking errors in various environments. The SLR algorithm demonstrates superior velocity tracking capabilities compared to the previous algorithm in most cases. Top performances are highlighted in bold. Note: *N/A* indicates that the ROA code repository does not support rapid locomotion.

Based on the evaluation results of the five code repositories, the proposed SLR algorithm generally achieves higher mean rewards and superior velocity tracking capabilities compared to the original implementations. From these results, we can draw the following conclusions:

1. The SLR algorithm achieves mean rewards comparable to those of the recently released top-performing HIM algorithm, but with a higher terrain level (see Figure 8), indicating superior performance. This demonstrates that learning from limited proprioceptive data can rival advanced privileged learning methods.

2. The SLR algorithm's effectiveness in managing complex tasks with MoB (multiple commands) and ROA (manipulator-equipped) repositories suggests that self-learning latent representations offer more general benefits than manually selected privileged information.

3. The SLR algorithm demonstrates enhanced tracking capabilities across various velocity ranges. One reason for this is that SLR can implicitly infer velocity from proprioceptive data. Additionally, the Critic network optimizes for maximum cumulative rewards, with velocity tracking as the most significant term in the reward function. Consequently, the backpropagation in the Critic network naturally guides the latent representation to optimize tracking performance.

### 5.3 Deploy in Real-World

The trained policies are deployed on the Unitree Go2 robot in real-world as depicted in Figure 7. Performance evaluation of the policy was conducted across various indoor and outdoor terrains, and comparative analyses were performed against code repositories [13, 27, 29] and Unitree built-in MPC control method. Each environment was tested 10 times. As presented in Table 2, the results demonstrate the superior efficacy of our policy in real-world scenarios.

# 6 Discussion and Limitations

In this study, we present a quadruped locomotion algorithm that operates without privileged information. Relying solely on limited proprioceptive data, the algorithm achieves SOTA performance. Looking ahead, we anticipate that future self-learning approaches will be able to integrate privileged information and external perceptions, leading to even more impressive outcomes.

While our blind policy demonstrates robust motion, achieving superior trajectory planning necessitates the use of vision sensors. Moving forward, we aim to enhance quadruped locomotion by integrating visual information for tackling even more complex challenges.

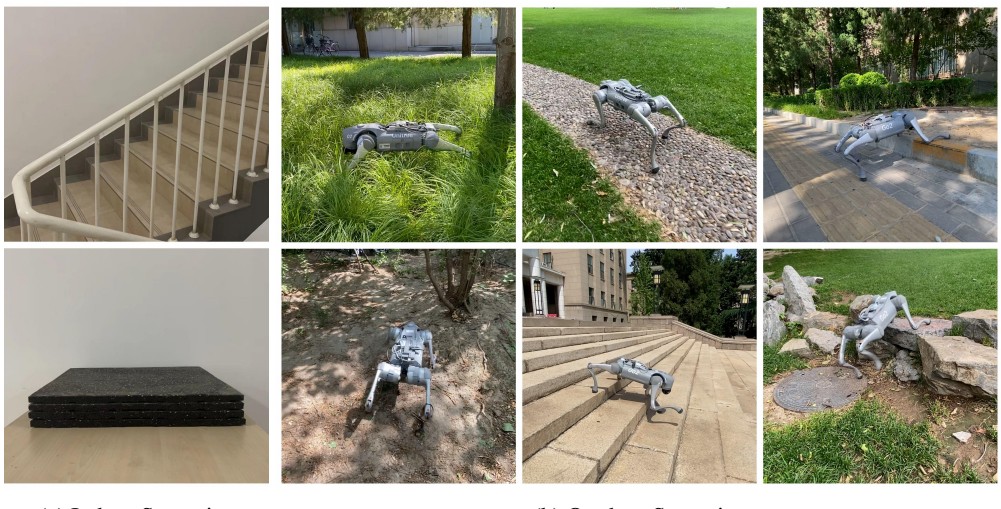

(a) Indoor Scenarios                    (b) Outdoor Scenarios

Figure 7: Indoor and outdoor experiment settings. The left column (a) showcases indoor scenarios, with the top row depicting the long stairs and the bottom row featuring the step obstacle. The right column (b) illustrates outdoor scenarios, depicting the robot moving through a bush, on a cobbled road, and climbing a curb in the top row, while the bottom row showcases the robot encountering a slope, stairs, and a rock.

| Scenarios | Terrain | Metrics | Ours | HIM | MoB | Baseline | MPC |
|---|---|---|---|---|---|---|---|
| Indoor | Stairs | Number | $\mathbf{136.4}^{\pm \mathbf{24.5}}$ | $107.4^{\pm 19.1}$ | $0.0^{\pm 0.0}$ | $0.0^{\pm 0.0}$ | $93.5^{\pm 32.9}$ |
|  | Step | Height (cm) | $\mathbf{35.4}^{\pm \mathbf{2.2}}$ | $30.7^{\pm 2.7}$ | $6.4^{\pm 1.2}$ | $5.3^{\pm 0.5}$ | $15.6^{\pm 1.6}$ |
| Outdoor | Bush |  | **100** | **100** | 90 | 50 | **100** |
|  | Cobbled Road |  | **100** | **100** | **100** | 30 | 90 |
|  | Curb | Success | **100** | **100** | 20 | 0 | 40 |
|  | Earthen Slope | rate (%) | **100** | **100** | 40 | 0 | 60 |
|  | Stairs |  | **100** | **100** | 0 | 0 | 20 |
|  | Rock |  | **70** | 30 | 0 | 0 | 0 |

Table 2: Comparison of deployment methods across various terrains. Each method was tested 10 times per terrain. The "Number " indicates the total stairs ($16 \times 29$ cm) climbed by the robot from start to fall, and "Height" shows the maximum step height the robot can handle. The $\pm$ symbol represents a 95% confidence interval. A successful trial means the robot completes the terrain without intervention or failure within the allotted time. Top performances are in bold.

**Acknowledgments**

We would like to express our gratitude to Han Yu for providing invaluable assistance in our deployment work. Additionally, this work is supported by the National Natural Science Foundation of China (No.U20A20220).

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

# A  Appendix

## A.1  Self-learning Latent Representation Details

---

**Algorithm 1** Self-learning Latent Representation

---

1: Randomly initialize adaptation module $\phi$, transition model $\mu$, and policy $\pi$
2: Initialize empty replay buffer $D$
3: **for** $itr = 1, 2, \ldots$ **do**
4:     $o_0 \leftarrow$ env.reset()
5:     **for** $t = 0, 1, \ldots, T$ **do**
6:         $z_t \leftarrow \phi(o_t^H)$
7:         $a_t \leftarrow \pi(o_t, \text{sg}[z_t])$
8:         $o_{t+1}, r_t \leftarrow$ env.step($a_t$)
9:         Store $(o_t, a_t, r_t, o_{t+1})$ in $D$
10:     **end for**
11:     $z_{t+1} \leftarrow \phi(o_{t+1}^H), \;\; \tilde{z}_{t+1} \leftarrow \mu(z_t, a_t), \;\; z_{t+n} \leftarrow$ RandomSample($D$)
12:     Compute $\mathcal{L}_{\text{trip}}$ (Eq.5) and $\mathcal{L}_{\text{ppo}}$ [24]
13:     Update $\pi, \mu, \phi$ by optimizing $\mathcal{L}_{\text{ppo}} + \alpha \mathcal{L}_{\text{trip}}$
14:     Empty $D$
15: **end for**

---

We presented the details of the Self-learning Latent Representation in Section 3.2 of the main paper. We set $H$ to 10 and the triplet coefficient $\alpha$ to 1.0.

## A.2  Reward Terms Detail

In Table A1, $v$ is the linear velocity, $\sigma$ is the tracking shaping scale equal to 0.25 here, $h^{\text{target}}$ is the desired base height corresponding to ground, $p_z^{\text{target}}$ and $p_z^i$ are the desired feet position and real feet position in z-axis of robot's frame and $v_{xy}^i$ is the feet velocity in xy-plane of robot's frame.

| Reward | Equation | Weight |
|---|---|---|
| Powers | $\|\tau\|\|\dot{\theta}\|^T$ | -2e-5 |
| Linear velocity tracking | $\exp\left\{-\frac{\|v_{xy}^{\text{cmd}}-v_{xy}\|_2^2}{\sigma}\right\}$ | 1.0 |
| Angular velocity tracking | $\exp\left\{-\frac{\left(\omega_{\text{yaw}}^{\text{cmd}}-\omega_{\text{yaw}}\right)^2}{\sigma}\right\}$ | 0.5 |
| Linear velocity penalty in z-axis | $v_z^2$ | -2.0 |
| Angular velocity penalty | $\|\omega_{xy}\|_2^2$ | -0.05 |
| Joint acceleration penalty | $-\|\ddot{\theta}\|^2$ | -2.5e-7 |
| Base Height penalty | $(h^{\text{target}} - h)^2$ | -10.0 |
| Joint torques | $-\|\tau\|^2$ | 1 |
| Action rate | $\|a_t - a_{t-1}\|_2^2$ | -0.01 |
| Action smoothness | $\|a_t - 2a_{t-1} + a_{t-2}\|_2^2$ | -0.01 |
| Foot clearance | $\sum_{i=0}^{3}\left(p_z^{\text{target}} - p_z^i\right)^2 \cdot v_{xy}^i$ | -0.01 |
| Orientation | $\|g\|_2^2$ | -0.2 |

Table A1: Reward Terms

### A.3 Domain Randomizations

| Parameters | Range[Min,Max] | Unit |
|---|---|---|
| Body Mass | [0.8,1.2]×nominal value | Kg |
| Link Mass | [0.8,1.2]×nominal value | Kg |
| CoM | [-0.1,0.1]×[-0.1,0.1]×[-0.1,0.1] | m |
| Payload Mass | [-1,3] | Kg |
| Ground Friction | [0.2,2.75] | - |
| Ground Restitution | [0.0,1.0] | - |
| Motor Strength | [0.8,1.2]×motor torque | Nm |
| Joint $K_p$ | [0.8,1.2]×20 | - |
| Joint $K_d$ | [0.8,1.2]×0.5 | - |
| Initial Joint Positions | [0.5,1.5]×nominal value | rad |
| System Delay | $[0,3\Delta_t]$ | s |
| External Force | [-30,30]×[-30,30]×[-30,30] | N |

Table A2: Domain Randomizations and their Respective Range

### A.4 Network Architecture

| Module | Inputs | Hidden Layers | Outputs |
|---|---|---|---|
| Encoder | $o_H$ | [256, 128] | $z_t$ |
| Actor | $o_t^H, z_t$ | [512, 256, 128] | $a_t$ |
| Critic | $o_t^H, z_t$ | [512, 256, 128] | $v_t$ |
| TransModel | $z_t, a_t$ | [256, 128] | $\tilde{z}_{t+1}$ |

Table A3: Network Architecture

### A.5 Hyper Parameters for Training

| Hyperparameter | Value |
|---|---|
| Clip range | 0.2 |
| Entropy coefficient | 0.01 |
| Discount factor | 0.99 |
| GAE discount factor | 0.95 |
| Desired KL-divergence | 0.01 |
| Learning rate | 1e-3 |
| Adam epsilon | 1e-8 |
| Replay Buffer Size | 4096×24 |
| Triplet loss coefficient | 1.0 |

Table A4: Hyper Parameters for Training

## A.6  Terrain Level Training Curves

In the Baseline [27] and HIM [29] code repositories, the default configuration includes a terrain curriculum. Consequently, some agents may progress to harder terrains, potentially resulting in lower returns. To verify the proposed algorithm's actual performance, terrain level training curves have been added, as shown in Figure 8.

Considering Figure 6 and Figure 8 together, it is evident that the actual performance of the SLR algorithm surpasses both of these methods.

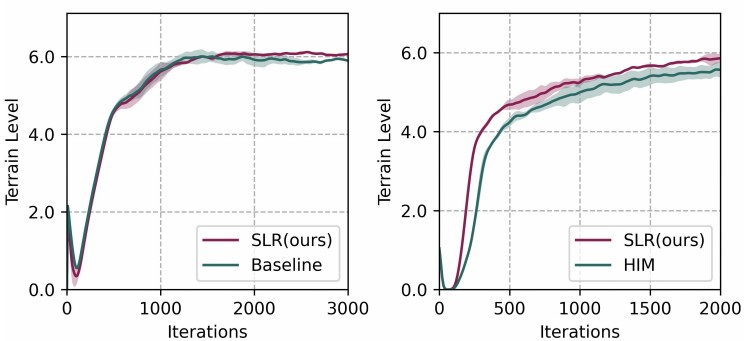

Figure 8: Terrain level training curves for Baseline and HIM code repositories.

