# OpenReview forum: "SLR: Learning Quadruped Locomotion without Privileged Information"
_robot-learning.org/CoRL/2024/Conference — CoRL 2024_

### Official Review · Reviewer_b8HJ · 2024-06-28

**Originality:** 3
**Technical Quality:** 3
**Clarity Of Presentation:** 3
**Potential Impact:** 3
**Recommendation:** 2
**Confidence:** 3

**Review:**

The paper proposes a robot locomotion learning framework that does not require privilege information during training stage and can adapt to different environment settings during deployment. It is achieved using a self-learning mechanism which learns to map different terrains to structured latent space. The results have been show in both simulation and real-world. The proposed method outperforms baseline methods in different evaluation metrics.

In the initial version of the paper, the conclusion that the method is useful is not well-supported by the numbers in the paper. The quantitative differences between the proposed method and baselines are either very limited or hard to judge. And the paper was also lack of explanation and insight to help understand the overall contribution of the paper.

In the rebuttal stage, the author provide more evidence to support the results, more result analysis is also provided. Therefore, compare to the initial version, the quality of the paper improves a lot.

**Quality Of The Limitations Section:**

3

**Questions For Rebuttal:**

I have no question for rebuttal.

**Robotics Focus:**

4

**Summary Of Paper:**

This paper aims to propose a quadrupedal locomotion learning framework that does not need privileged information during training stage.

**Summary Of Recommendation:**

The paper is more convincing as new results and explanation provided during rebuttal.

---

### Official Review · Reviewer_Bcyx · 2024-07-20

**Originality:** 3
**Technical Quality:** 3
**Clarity Of Presentation:** 4
**Potential Impact:** 3
**Recommendation:** 3
**Confidence:** 4

**Review:**

The paper presents an interesting method and is generally easy to follow. It provides sufficient coverage of related work. The authors have also clearly delineated the position of the paper.

Strengths
- The learned latents can differentiate between different terrains with the t-SNE visualizations depicting distinct ring-like structures.
- The proposed algorithm has been reimplemented in the repositories of the baseline environments without affecting the other simulation parameters to ensure fair comparison.
- The authors demonstrate impressive real-world performance, with the robot able to walk on various terrains and robust to external disturbances.

Weaknesses
- The proposed method section could benefit from additional details about the setup. This will help with clarity and reproducibility.
- The terrain transitions argument can be substantiated further, especially since this effect is being validated in sim-only. The authors can potentially test a few other terrain sequences to obtain a clearer picture of how their self-learned latent represents transitions.

**Quality Of The Limitations Section:**

2

**Questions For Rebuttal:**

* It would be great if the authors could provide more details about the training process for triplet optimization. For example, it is unclear how the other latents are sampled to enforce distinctiveness, or what the frequency of updates for the encoder is. It might be nice to explicitly show the overall sequence in the form of pseudocode or an algorithm. This will help with reproducibility.
* Given that some of the papers cited by the authors report more more dynamic behaviors (such as Extreme Parkour and Rapid locomotion via reinforcement learning), it might be nice to have higher target velocity ranges, especially for comparison against the MoB and RLvRL baselines which seem to have better linear velocity tracking but worse angular velocity tracking than the proposed method.
* Another minor suggestion is to cite the respective paper with the baseline in section 5.1 as this will further help improve clarity.
* While the real-world deployment performance is impressive, it would further help to explicitly specify the success criteria and relevant metrics (such as distance covered before a fall or time before a fall etc) for further transparency.

**Robotics Focus:**

4

**Summary Of Paper:**

Incorporating privileged information to learn a latent vector encoding various environmental parameters or using it to estimate hard-to-compute parameters have proven to be very effective techniques in the domain of quadrupedal locomotion. The given paper operates in the blind locomotion setting and tries to estimate these environmental parameters without explicitly using privileged information. The proposed method instead relies on using just proprioceptive information to encode the observation history into a latent embedding and learning a state transition estimator for the latent. This latent embedding allows the learned policy to operate in different terrains and transition smoothly between them, and be robust to external disturbances. The authors term this the Self-learning Latent Representation (SLR) method.

**Summary Of Recommendation:**

While the authors show improvement in performance compared to the baselines, the main differentiating factor lies in the interpretability of the latent. It might be interesting to extend it to more dynamic and agile behaviors, especially trying to transition between such motions. The paper could be improved further by providing additional details about the implementation of the proposed method and a slightly more thorough substantiation of the claims.

---

### Official Review · Reviewer_fW7r · 2024-07-21
**Review of Self-Learning Representations for Quadruped Locomotion without Privileged Information**

**Originality:** 4
**Technical Quality:** 4
**Clarity Of Presentation:** 4
**Potential Impact:** 4
**Recommendation:** 3
**Confidence:** 5

**Review:**

This research work introduces a novel and simplified end-to-end learning-based locomotion method, which autonomously learns the latent representation without the need for laborious teacher-student privileged information learning-based methods. This approach enhances the fairness of evaluation by comparing against SOTA methods within their environment configurations. The concept of self-learning latent representations is well-motivated and holds significant importance in the field of learning-based locomotion. The paper clearly and comprehensively presents the proposed framework, demonstrating strong clarity and organization. Furthermore, the extensive real-world demonstrations substantiate the high quality and practical applicability of this research, providing robust evidence of its effectiveness and reliability.

**Strengths**

1. **Easiness of training method**: The self-learning latent representation (SLR) method simplifies the training process by eliminating the need for selecting and experimenting with combinations of privileged information, as required in current SOTA methods [10], [12], [13] mentioned in the paper.
2. **Fair Evaluation**: Implementing SLR within SOTA open-source code repositories and comparing the reference using their optimized configurations ensures a fair evaluation. This approach demonstrates the effectiveness of SLR under the same conditions as existing methods, providing a reliable comparison of performance.
3. **Clear Ablation Study**: The t-SNE visualizations presented by the authors show that SLR latent representations have a strong correlation with the underlying terrains. The visualizations depict upslope and stair climbing shares similar understanding of the environment, effectively illustrating the method's ability to capture and differentiate the nuances of various terrains.
4. **Strong Real-World Experiments**: The submitted video shows successful and robust behaviors in indoor and outdoor environments

**Weakness**
1. **Limited Analysis on Linear Velocity Tracking**: In comparison with four reference SOTA algorithms shown in Table 1, the authors didn't provide sufficient evidence explaining why the linear velocity tracking (LVT) is underperforming compared to angular velocity tracking (AVT) in the case of SLR. This lack of analysis leaves a gap in understanding the specific challenges and limitations of SLR in LVT scenarios.
2. **Power Consumption Comparison**: The authors didn't compare the overall power drawn by their policy against other methods. Observations from the submitted video (00:35) suggest that the robot tends to raise its forelegs higher even on cobbled roads, which might lead to higher power consumption. However, this could potentially be addressed by incorporating more penalty for power consumption in the reward function.

**Quality Of The Limitations Section:**

3

**Questions For Rebuttal:**

**Issues**

1. In Section 3.2, please provide more details of the neural networks used in your framework, either in the manuscript or supplementary material. Specifically, what is the dimension of 𝑧𝑡 ? What is the size of the MLP for the encoder and the state transition model?
In line 112, please clarify how the state transition model and encoder share the same network structures despite having different inputs.
2. What is the size of the Replay Buffer?
3. In the loss function, what is the value of the margin 'm' used?
4. In Section 5 (Results), please consider citing the acronyms of reference methods for clarity.
5. What was the reason that RMA [10] was not considered a potential reference for comparison? While RLvRL [9] uses RMA to train its policy, [9] is more biased towards rapid locomotion tasks. It would be interesting to see a performance comparison against RMA [10] as it has also shown similar robust behaviors in indoor and outdoor environments.

[9] G. B. Margolis, G. Yang, K. Paigwar, T. Chen, and P. Agrawal. Rapid locomotion via reinforcement learning. The International Journal of Robotics Research, 43(4):572–587, 2024.

[10] A. Kumar, Z. Fu, D. Pathak, and J. Malik. Rma: Rapid motor adaptation for legged robots. 2021.

**Robotics Focus:**

4

**Summary Of Paper:**

This paper proposes a Self-learning Latent Representation (SLR) method for robust quadrupedal locomotion over complex terrains. Unlike state-of-the-art (SOTA) learning-based methods, which rely on meticulously chosen privileged information in simulations to learn a latent representation of the physical properties of the environment, SLR self-learns these latent representations guided by state-transitions, state distinctions, and cumulative rewards under a Markov Decision Process (MDP). The authors demonstrate successful deployment of the learned policy both in simulation and on real robots across various complex terrains. Additionally, they compare the effectiveness of the SLR method against four SOTA open-source references, highlighting its robustness and adaptability in real-world applications.

**Summary Of Recommendation:**

I would like to recommend this paper for acceptance. This paper presents a significant and original contribution to the field of learning-based locomotion for quadrupedal robots. The novel approach of self-learning latent representations without relying on privileged information simplifies the training process and demonstrates robust performance across various complex terrains. While there are areas for improvement as mentioned in the weakness and rebuttal section, the strengths of this work make it a valuable addition to the research community.

---

### Official Review · Reviewer_ztJ4 · 2024-07-26

**Originality:** 3
**Technical Quality:** 3
**Clarity Of Presentation:** 3
**Potential Impact:** 3
**Recommendation:** 3
**Confidence:** 5

**Review:**

The paper proposes SLR, a representation learning technique for RL locomotion based on triplet optimization.

Strengths:
- SLR learns a better internal representation than some prior works without relying on privileged information. Avoiding privileged information makes the technique more general because it doesn't rely on the designer to come up with a good choice of privileged variables and compute them during training. Overall, I really like the idea of incorporating contrastive learning in this way during training.
- SLR consistently improves training in open-source codebases from diverse sources
- The visualization of latent space reveals a better clustering of latents when the robot is walking on the same or similar terrains.

Weaknesses:
- Some of the metrics do not fully evaluate the performance difference between SLR and baselines evaluated on the same environment, because they report training return in environments with performance-adaptive curriculum learning.
- Some tables should be reformatted for clarity
- More baseline comparisons against recent works would strengthen the paper

**Quality Of The Limitations Section:**

3

**Questions For Rebuttal:**

- Figure 6 reports the reward curves for various repositories. Is this the reward from the training rollouts or in a fixed evaluation environment? Some of these environments have a performance-dependent curriculum, e.g. on velocity or terrain, so that a higher training reward does not necessarily reflect higher performance, since the robot might be training on easier terrains or tasks. I believe Table 1 provides a fair comparison evaluating Author vs SLR on a fixed evaluation environment which would be correct, but this should be explicitly discussed. The authors should add some discussion on this to indicate what can be interpreted from each result.
- Similarly, Figure 3 reports reward curves that I expect are based on training returns; but my understanding of [27] is that there is a terrain curriculum, where some policies may advance to significantly harder terrains, and might also be responsible for a lower return. The authors should add (1) plot of the terrain level progression during training, and (2) evaluation of the policies on a fixed evaluation environment (e.g. all level 5 terrains)
- Table 2 shows that SLR can traverse terrains well, but it should be noted that the MoB and RLvRL methods are trained on flat ground. It would be more straightforward to emphasize the performance of ablations of SLR (SLR w/ explicit privileged latent, SLR w/ implicit privileged latent, SLR w/o latent) in one table and the performance vs. other controllers (MoB, RLvRL, MPC) in a separate table, and I would find the ablation table much more interesting. The naming of "Baseline" is not descriptive and is also placed far from SLR which makes it harder to interpret what I want to know from the current table.
- Comparison with DreamWaq would strengthen the paper, since it also claims to learn a better latent representation for locomotion; but I know there is no official code release so this might take more work to provide.
- https://arxiv.org/abs/2312.11460 also proposes a contrastive-based approach to legged locomotion. On one hand, the code was only released after the CoRL submission deadline so this might be considered a concurrent work. But I think comparing HIM vs SLR would strengthen the impact of the paper because I would know which contrastive technique yields better performance.

**Robotics Focus:**

4

**Summary Of Paper:**

This paper proposes SLR, a representation learning technique for RL locomotion based on triplet optimization.

**Summary Of Recommendation:**

I like the idea of contrastive learning to form a better latent representation during RL training. The authors should clarify the significance of comparing training reward curves in these environments and revise the results for clarity. Comparison to a recent/concurrent related work would strengthen the paper's impact.

---

### Author Rebuttal · Authors · 2024-08-09

Thank you for the valuable feedback from the Area Chair and all reviewers. We have carefully revised the manuscript, with changes highlighted in the text. The main revisions are:

1. Adding ablation experiments (SLR with explicit, SLR with implicit, SLR without latent).
2. Supplementing Terrain Level Training Curves.
3. Including a comparison with the recently released top-performing HIM algorithm.
4. Expanding the velocity tracking evaluation experiments.
5. Providing detailed conclusions for each set of experiments.
6. Optimizing deployment experiments.
7. Highlighting the innovations and potential of the proposed method in key sections of the paper.

We appreciate your suggestions.

---

### Decision · Program_Chairs · 2024-09-04

**Decision:**

Accept

**Comment:**

I agree with the majority of the reviewers and recommend the acceptance of the paper. The reviewers are mainly positive about the paper (3 out of 4 weak accept). The main strength of the paper are
* the simplicity of the algorithm which does not rely on privileged information or student-teacher methods
* the fair evaluation of the algorithms within the original environments
* the real-world experiments of indoor and outdoor environments

I want to especially highlight the fairness of the evaluation as the author used the original environments of the previously published works and evaluated their methods under their conditions. The main weakness of the paper is the limited (potentially marginal) improvement over prior art and the debatable novelty as pointed out by reviewer b8HJ, who was initially not very constructive but improved during the rebuttal. While one could argue in either direction for novelty, I personally would consider the work novel as the exact combination has not been shown in the literature!

I recommend accepting the paper because it proposes a simple algorithm that works on the physical system and the paper provides a fair comparison to prior art. I am very certain in my evaluation as the paper was reviewed by many reviewers that extensively published on the topic.